# Promoting Influenza Vaccination Uptake Among Chinese Older Adults Based on Information–Motivation–Behavioral Skills Model and Conditional Economic Incentive: Protocol for Randomized Controlled Trial

**DOI:** 10.3390/healthcare12232361

**Published:** 2024-11-25

**Authors:** Hao Lin, Jiannan Xu, Refukaitijiang Abuduwayiti, Ying Ji, Yuhui Shi, Lanchao Zhang, Zhengli Shi, Mojun Ni, Sihong Tao, Bohao Yang, Shuhan Liu, Omar Galárraga, Chun Chang, Wangnan Cao, Phoenix Kit-Han Mo

**Affiliations:** 1School of Public Health, Peking University Health Science Center, Beijing 100191, China; linhao@bjmu.edu.cn (H.L.); 1710306113@pku.edu.cn (J.X.); 1910306243@pku.edu.cn (R.A.); jiying@bjmu.edu.cn (Y.J.); yuhuishibmu@bjmu.edu.cn (Y.S.); 2311110238@pku.edu.cn (L.Z.); zhenglishi@stu.pku.edu.cn (Z.S.); nmj2020@bjmu.edu.cn (M.N.); 2411210089@bjmu.edu.cn (S.T.); 2210306225@stu.pku.edu.cn (B.Y.); 2210306230@stu.pku.edu.cn (S.L.); 2Center for Healthy Aging, Peking University Health Science Center, Beijing 100191, China; 3Department of Health Services, Policy and Practice, Brown University School of Public Health, Providence, RI 02903, USA; omar_galarraga@brown.edu; 4School of Public Health and Primary Care, The Chinese University of Hong Kong, Hong Kong, China; phoenix.mo@cuhk.edu.hk

**Keywords:** influenza vaccination, older adults, information motivation behavioral skill model, economic incentive, randomized controlled trial, China

## Abstract

**Background**: Influenza poses a substantial health burden, especially among older adults in China. While vaccination is one of the most effective preventions, influenza vaccine uptake rates among Chinese older adults remain low. This study examines the individual and combined effects of behavioral interventions based on the Information–Motivation–Behavioral Skills (IMB) model and economic incentives in promoting influenza vaccine uptake among older adults living in China. **Methods**: The study will recruit 640 older adults living in eight communities that have not been covered by the free influenza vaccination policy. These eight communities (as clusters), stratified by urban and rural, will be randomized to four parallel arms, including a usual care arm, an IMB-based behavioral intervention arm, a conditional economic incentive arm, and a combined behavioral-economic arm. The interventions will start from the beginning of the flu season and last for about one month. Specifically, the IMB-based behavioral intervention encompasses health education brochures, healthcare provider-led lectures, interactive quizzes, and personalized consultations. The exact number of economic incentives is conditional on the timing of vaccination (a higher amount for early immunization) and the number of people within a household to be vaccinated at the same appointment (a higher amount for more people). The primary outcome is the influenza vaccination rate. Data will be gathered through vaccination records and questionnaires covering IMB-based vaccination cognitions. Mixed-effects models will be used to analyze the outcome of vaccination rate, reporting difference-in-differences estimates with 95% confidence intervals. **Conclusions**: The results of this study have the potential to inform influenza vaccination program scaleup among older adults who are not yet covered by the free influenza vaccination policy. **Ethics and dissemination**: Ethics approval has been granted by the ethics commission of Peking University Health Science Centre (IRB00001052-24090). Participants will be required to sign a written consent form. Findings will be reported in conferences and peer-reviewed publications in accordance with the recommendations of the Consolidated Standards of Reporting Trials. **Registration number**: This study was registered at the Chinese Clinical Trial Registry (ChiCTR2400090229).

## 1. Introduction

Influenza annually impacts a substantial fraction of the world’s population, causing significant health, social, and economic consequences. Globally, influenza causes an estimated 3 to 5 million cases of severe illness and 290,000 to 650,000 respiratory deaths annually [1]. Older people are generally more vulnerable to influenza. Research indicates that over 85% of influenza-related deaths occur among individuals aged 65 and above [2]. In mainland China, studies have estimated influenza-associated all-cause mortality rates of 14.33 per 100,000 persons for all ages and 122.79 per 100,000 persons for those aged ≥ 65 years [2]. This is accompanied by a significant economic burden among older people infected with influenza [3]. Research indicates that the average hospitalization cost for older adults due to influenza—around CNY 14,000–19,000 (approx. USD 1929–2618)—is double that of non-older adults, equating to roughly 60% of the average annual income of urban older adults and twice that of rural older adults [4].

Vaccination is the most cost-effective means of combating influenza [5]. Studies have shown that the influenza vaccine, aimed at protecting high-risk groups from severe influenza cases, effectively reduced hospitalizations for pneumonia or influenza by 27–60% and lowered the mortality rate by 48–80% in community-dwelling older adults [6,7,8]. The WHO recommendations underscore the necessity of annual vaccination for older adults aged 65 and above, urging/calling upon its Member States to achieve at least 75% vaccination coverage among older adults by 2010 [9]. In most regions of China, the influenza vaccine is not part of the national broad immunization program, and individuals must pay the market price (about USD 20) to get the vaccination. Despite the significant geographical variations, the overall rate of influenza vaccination of older adults remains low across the country, ranging from 33% to 46% in regions offering free vaccination and 0.5% to 5% in regions without free vaccination [10].

Different levels of factors were found to be associated with influenza vaccination, including socio-demographics (age, education, occupation, health insurance coverage, income, living status, impaired functional status), health service-related factors (vaccination costs, recommendation from a physician, accessibility to healthcare services) and cognitive factors (vaccination awareness, low vaccine literacy, concerns about the vaccine’s side effects) [11,12,13,14,15].

To address these factors, various behavioral interventions aimed at increasing influenza vaccinations have been documented internationally and locally in China. Strategies to improve vaccination rates among older people have focused on enhancing access to vaccinations, addressing concerns about costs and side effects, and boosting awareness within communities [16]. Specific interventions have included sending reminders about the benefits of vaccination [17], providing health education messages [18,19], conducting home visits [19], and utilizing pharmacy-based vaccination delivery [20]. These approaches specifically target the barriers identified, such as limited access and lack of awareness, while also leveraging positive influences, such as community demand and healthcare provider recommendations.

Evidence suggests that interventions grounded in a theoretical framework tend to be more likely to succeed in increasing influenza vaccination rates [21]; theoretical frameworks commonly reported include the Health Belief Model (HBM), the Theory of Reasoned Action (TRA), and the Health Action Process Approach (HAPA) [22]. Among these, the Information–Motivation–Behavioral Skills (IMB) theory has been particularly influential in guiding the design of interventions to explain vaccine uptake behavior. The IMB theory posits that information, motivation, and behavioral skills are critical in influencing health behaviors. Information refers to an individual’s understanding of the vaccine, its benefits, and the risks associated with influenza. Motivation encompasses personal attitudes towards vaccination and social influences like norms and peer support. Finally, behavioral skills involve the practical abilities required to obtain the vaccine, such as accessing healthcare services and overcoming logistical barriers [23]. Research has shown that the low influenza vaccine uptake rates among Chinese older adults are highly related to low information, low motivation, and low behavioral skills [12,24], indicating that the IMB model has great potential to guide interventions to improve this vaccine uptake rate. Relevant constructs tested in previous interventions included vaccine knowledge, risk perception, beliefs about the vaccination benefits, subjective norms related to vaccination, and cues to action [12]. Additionally, existing studies that utilized the IMB framework have found that most of its elements were significantly associated with vaccine uptake intention and behavior [25].

Despite the popularity of theory-based interventions, the effectiveness of the interventions above had been mixed [16], and cross-study comparison was difficult because interventions varied depending on the setting and participants’ backgrounds. Third, behavioral interventions conducted in the Chinese setting are relatively limited, and very few studies have taken the urban-rural differences into consideration.

In addition to behavioral interventions, economic incentive-based interventions have emerged as another useful approach to bolster influenza vaccination rates [23]. Prior research underscores that economic incentives promoting health-conducive behaviors can stimulate behavioral shifts [25,26]. A pragmatic randomized controlled trial in Beijing, China, investigated the impact of financial incentives ranging from CNY 0 to 60 on influenza vaccine uptake among older adults (≥60 years). The findings showed that providing economic incentives boosted influenza vaccine uptake (41% with an amount of CNY 60 vs 37% without incentives) [27], and similar results have been observed in studies conducted in Singapore [28]. Incentives can boost vaccine uptake by lowering access barriers and enhancing motivation. They may include financial rewards or community recognition, which help offset costs and encourage individuals to seek vaccination. A quasi-experimental pragmatic trial evaluated the effectiveness of a pay-it-forward intervention in increasing influenza vaccine uptake among older adults in China, and the results were promising, with 60% in the pay-it-forward arm and 20% in the usual care arm [29]. The pay-it-forward intervention encouraged vaccinated individuals to support others in their community, creating a cycle of reciprocity. For instance, those who are vaccinated might contribute to a fund to help others get vaccinated. This approach not only increases individual uptake but also fosters community solidarity around health [29].

All of the economic interventions above were unconditional, meaning the participants received the incentive regardless of their vaccination behaviors. Conditional incentive and unconditional incentive may produce different outcomes in terms of intervention effect and cost-effectiveness [30]. A conditional incentive means that participants are given different choices, and they may receive different amounts of incentives based on the participant’s performance. Research indicates that community engagement and neighborhood social cohesion are important targets for promoting health behaviors among older adults [31,32]. These findings suggest that conditional economic incentive approaches that leverage social norms and family support can be a promising strategy for boosting vaccination rates among older adults.

Despite existing research on factors influencing older adults’ willingness to receive influenza vaccines and behavioral interventions, several knowledge gaps remain. First, influenza vaccine intervention programs often lack a solid theoretical foundation in health behavior, resulting in suboptimal intervention effects. Second, considering the prevalent practice of self-paid influenza vaccination in most parts of China, there is a pressing need for rigorous research to comprehensively evaluate the impact of economic incentive interventions on vaccination willingness and uptake, particularly in the Chinese setting, as the national program does not cover influenza vaccination.

The present study aims to test the effectiveness of IMB-based behavioral intervention and conditional economic incentives in increasing influenza vaccination among older adults in China through a randomized controlled trial.

## 2. Materials and Methods

### 2.1. Study Design

This study will use a cluster randomized controlled trial design, with the community as the unit of randomization and various interventions, to evaluate the effectiveness of IMB-based behavioral intervention and conditional economic incentives among the Chinese older adults. The primary outcome is influenza vaccination over the follow-up durations.

We will choose communities from different provinces where the influenza vaccine is not covered by the present health system and older adults needs to pay out of pocket. In the two targeted provinces (Jiangsu and Shandong), four communities will be selected from urban and rural areas within each province, resulting in eight communities in total. These eight communities will be stratified based on their urban or rural status and then randomly allocated to (1) the IMB-based behavioral intervention group, (2) the conditional economic incentives group, (3) the combined IMB-based behavioral intervention and conditional economic incentives group, and (4) the control group. We will assess influenza vaccine uptake before and after the intervention and between different study arms. Figure 1 illustrates the study design.

### 2.2. Study Population and Participant Recruitment

The eligibility criteria for older adults in this study are as follows: (a) age 60 years or older; (b) residing in the community for at least six months with no plans to move out; (c) one participant in each household; and (d) voluntarily participating in the study and able to provide informed consent. Those who meet any of the following conditions are excluded: (a) having a physical health condition deemed unsuitable for influenza vaccination; (b) having a cognitive impairment or other severe illness that may affect the informed consent process.

In this study, we will collaborate with community health service centers to recruit study participants and carry out the project. Community collaborators will participate in the project training, assist with participant recruitment, support baseline and follow-up surveys, and assist in quality control.

### 2.3. Sample Size and Sampling

The primary outcome in this study is the influenza vaccination rate among older adults, assuming a two-sided test, a type I error rate (α) of 0.05, and a type II error rate (β) of 0.10. Based on a literature review, the current median influenza vaccination rate among older adults is 8% in China at national level [10], and the target vaccination rate after the IMB-based behavioral intervention is set at 16%. Accounting for the cluster sampling approach (with a design effect between 1.3 and 1.5), the required sample size per group is 72, for a total sample size of 576 participants. Assuming a 10% loss to follow-up in the final assessment, the total sample size is 640 participants, with 160 participants per arm, across two provinces and eight communities.

The client list of Essential Public Health Services in each community health service center will be used to facilitate participant randomization and recruitment. Within each participating community, a simple random sampling approach will be used to recruit 80 participants using computer-generated random numbers. All screening and recruitment will be carried out by the local staff working at the community health service center.

### 2.4. Intervention

The intervention started before the influenza season, running from October this year to February next year. Before the start of the influenza season, the eight communities were randomly allocated into four study arms: (1) IMB-based behavioral intervention, (2) conditional economic incentives, (3) a combination of IMB-based behavioral intervention and conditional economic incentives, and (4) the control group with usual care, with stratification by urban and rural areas.

**IMB-based behavioral intervention.** The older adults will receive behavioral intervention based on the IMB model (Figure 2). The underlying idea of the intervention is that vaccination behavior is closely related to the older adults’ vaccine knowledge, their motivation to reduce the risk of influenza infection through vaccination, and vaccination-related behavioral skills. The IMB model includes the following components: (a) information, comprising objective and subjective vaccine information; (b) motivation, including perceived susceptibility, attitudes (towards vaccines), and complying with social norms (e.g., peer recognition of vaccine); and (c) behavioral skills, mainly encompassing vaccine-related self-efficacy (e.g., managing side effects), action planning (scheduling vaccinations), and coping planning (dealing with vaccination obstacles) [33]. The conditional economic incentives could lead to behavior change via potential pathways, involving modulating the price effect, enhancing the income effect, and mitigating high discounting or present bias [34].

The main components of the intervention, designed based on the IMB model, will be implemented offline and include the following:

(a) Distribution of health education brochures. The printed health education brochures summarizing official information on influenza vaccination from WHO and China’s National Health Commission will be distributed once after the baseline.

(b) Lectures on influenza vaccination. Influenza vaccination experts will lead the lectures. Topics will cover information about influenza vaccine, improving intentions for influenza vaccination uptake, social norms for vaccination, self-efficacy in vaccination and managing potential side effects, and creating effective action plans for vaccine appointments. The research team will provide lecture materials. This component will also take place once.

(c) Interactive quizzes on influenza knowledge with prizes. Participants will engage in quizzes on influenza knowledge, using questions sourced from official core information. This activity will occur once.

(d) One-on-one consultations with doctors. Participants will have access to one-on-one consultations with doctors during the intervention period.

The intervention program will be implemented after the baseline assessment. The interventions will start at the beginning of the flu season and last for about one month. Supported by the research team, the community staff will be responsible for delivering the intervention and monitoring the process.

**Conditional economic incentives.** Two incentive conditions have been established: one is based on the timing of vaccination within the flu season, and the other is based on the number of people vaccinated together. First, each participant vaccinated within the first month of the flu season receives CNY 30 (or about USD 4.25 at current rates) as the early-bird incentive. The incentive is CNY 15 if the individual gets vaccinated within the second month and CNY 0 after the second month. Second, each participant receives CNY 30 if three or more people within the same household are vaccinated on the same day, while it is CNY 15 for two people and CNY 0 for one person vaccinated alone. The above two conditional incentives can be combined, offering a maximum total incentive of CNY 60 per person, which amounts to approximately 50% of the current market price for the influenza vaccination in 2024, set at CNY 114. Participants will be informed of the incentive system upon entering this study. The incentives will be delivered in cash after the verification of vaccination records.

**A combination of IMB-based behavioral intervention and conditional economic incentives.** Participants will receive both the IMB-based behavioral intervention and the conditional economic incentives during the intervention.

**Control arm: usual care.** Participants will receive usual care without additional interventions. This includes access to routine healthcare services, general health information available within the community, and informational posters in public spaces to promote influenza vaccination among all community members.

### 2.5. Measurements

Participants will receive information about the study purpose, content, and procedures and sign a written informed consent form before the baseline assessment. Our research team independently developed the questionnaire regarding the scales used in the existing literature. The questionnaire includes the five main aspects: socio-demographic characteristics, vaccination intentions and status, vaccine literacy, and IMB-based vaccine cognization. A panel of specialists in epidemiology, psychology, and behavioral science will be invited to review and evaluate the face validity of the questionnaire and make the modification suggestions. The questionnaire will be tested on about 20 participants before formal use. Participants who complete the baseline and follow-up questionnaires will receive CNY 20 (≈USD 2.7) each time as compensation for their time.

**Sociodemographic characteristics:** Sociodemographic information will be collected, including gender, age, ethnicity, education level, occupation before retirement, place of residence, health status (including the presence of any chronic illnesses), personal monthly income, and insurance coverage status.

**Influenza vaccination intentions and status:** The vaccination intentions will be measured through questions assessing the participants’ willingness to receive the influenza vaccine in the current year. A five-point Likert scale ranging from “1 = strongly disagree” to “5 = strongly agree” will be used to capture their intention levels [35]. We will investigate the detailed information on the participants’ previous influenza vaccination experiences, encompassing their vaccination history in the last influenza season (from September 2023 to February 2024), the mode of payment for the vaccination, the accompanying individuals during the vaccination process, and any reasons cited for not being vaccinated. Information on pneumococcal vaccination status will also be collected.

**Influenza vaccine literacy:** Participants will answer questions about general vaccine literacy and influenza vaccine literacy. We will measure and evaluate both types of vaccine literacy through two dimensions: functional literacy and knowledge literacy. For the vaccine literacy measurements, the first draft was developed based on the World Health Organization (WHO) Vaccine Literacy Questionnaire, the Health Literacy Measurement Tool for Vaccines (HLS19-vac) and the China Influenza Vaccination Technical Guideline (2023–2024) [36,37,38]. Then the draft was reviewed by a multidisciplinary team to check for clarity and relevance to the Chinese context. The revised draft was finally pilot tested among 10 older individuals before its formal use.

For knowledge-based general vaccine literacy, we will investigate participants’ general knowledge about vaccines, such as their effects, side effects, and possible effects on the immune system. For functional general vaccine literacy, we will examine the behavioral ability of study participants to obtain information about vaccines, recognize the importance of vaccination, and determine the need for vaccination.

For the knowledge-based influenza vaccine literacy dimension, we will ask study participants about their knowledge of the benefits of influenza vaccination, physical reactions after vaccination, the relationship between the effectiveness of the influenza vaccine and influenza strains, and the preventive effects of the influenza vaccine. For functional influenza vaccine literacy, study participants will be asked about the recommended frequency of influenza vaccination, cost, perception of influenza prevalence, and ability to self-assess vaccination needs. We will also collect information on the perceived risks associated with influenza infections and the primary sources of vaccine-related details (e.g., healthcare providers, media, social networks) among participants.

**IMB-based vaccination cognitions:** To develop these vaccination-related cognition measurements, we first conducted a thorough literature review of existing surveys and measures related to vaccination cognitions, particularly those guided by IMB theory or similar constructs including information (knowledge), motivation (positive attitudes) and behavioral skills (perceived behavioral control, capability). Then, we formed an item pool and critically reviewed by a multidisciplinary team to ensure its relevance and clarity [39,40]. For the IMB-based vaccine cognition section, we surveyed the following aspects using a Likert five-point scale, ranging from 1 (strongly disagree) to 5 (strongly agree): perceived susceptibility of influenza (e.g., “contracting influenza will impact my physical health”), attitudes towards vaccines (e.g., “influenza vaccination can prevent me from getting influenza”), the intention of influenza vaccination (e.g., “I will get the vaccinated this flu season”), social norms (e.g., “family, friends, or peers suggested that I get the influenza vaccine”), vaccine-related self-efficacy (e.g., “I know how to manage the side effects after receiving the vaccine”), action planning (e.g., “I can find out the time and location for influenza vaccination”), and coping planning (e.g., “I am aware of the procedure of influenza vaccination”).

The follow-up survey will be conducted during the final month of the influenza season, which is February of the following year. This follow-up will encompass two key components: a questionnaire survey (identical to the baseline questionnaire) and vaccination status verification through official vaccination records maintained by healthcare providers at community health service centers.

### 2.6. Outcomes

The primary outcome is the influenza vaccination rate. The vaccination records will be collected from the collaborating health service center databases. This study does not provide free vaccination; participants need to pay for vaccination at the market price. The secondary outcome is the intention of influenza vaccination (under the free and paid scenarios), primarily from questions in the follow-up questionnaires.

### 2.7. Data Analysis

The mean and standard deviation will be used to describe the data for numerical variables such as intention to be vaccinated and scores on each element of the IMB model. The *t*-test will be used to compare the balance of baseline data between the intervention group and the control group for these variables. For categorical variables such as influenza vaccination, the rate will be used to describe the data, and the chi-square test will be used to compare the balance of baseline data between the intervention group and the control group.

We will estimate the Difference in Differences (DIDs) (with a 95% confidence interval (CI)) of the proportion of vaccine uptake between each intervention and control (non-intervention) arm, adjusting for the clustering effect and testing whether it is statistically significant or not at a 5% level of significance. This study will use mixed-effects models to evaluate the impact of the primary and secondary outcome indicators. A generalized mixed linear model will be used to assess the impact of categorical outcome variables, and a mixed linear model will be used to evaluate the effect of continuous outcome variables. The effect evaluation process will follow the intention-to-treat principle. All statistical analyses will be performed using IBM SPSS Statistics (version 27) and SAS (University Edition, version 9.4). The statistical significance level will be *p* < 0.05.

### 2.8. Data Management

Researchers will input data from participants’ original observation records into the study subject record forms, and the completed survey forms will be promptly submitted to the research data manager. Data entry will be performed using a dual-entry database system and will undergo two rounds of comparison. If any issues arise during this process, the researchers will be required to provide explanations. Once all subject record forms have been entered twice and verified without errors, the data manager will generate a database check report. This report will include information on study completion (including a list of dropouts), eligibility/exclusion criteria checks, integrity checks, logical consistency checks, outlier data checks, time window checks, and adverse event checks. The data files, including the database, validation programs, analysis programs, results, codebooks, and explanatory files, will be categorized and kept private using a unique identification number. Participants’ data privacy will be ensured, and all data collected throughout the research will be kept strictly confidential and will not be shared with anyone else. Also, there will be strong controls on who has access to the data forms and rigorous adherence to maintaining data confidentiality. There will be no sharing of participant names or any other identifying information.

### 2.9. Quality Control

Rigorous quality control measures will be implemented throughout the research process. Detailed work plans, implementation manuals, subject record forms, and informed consent forms will be developed and subjected to rigorous ethical review for approval. We will establish training and assessment programs for all research personnel to ensure a uniform understanding of inclusion/exclusion criteria, standardize data recording procedures, and maintain a consistent outcome assessment. A pre-study pilot test involving at least 40 participants will be conducted to refine study procedures and identify potential issues before the main study commences. Any participant who withdraws from the study before completing the prescribed observation period will be designated a dropout. Researchers will actively seek contact with dropout subjects to document reasons for the last intervention date and complete as many assessments as feasible. Additionally, all dropout subjects’ records will be retained for analysis.

## 3. Discussion

This randomized controlled trial protocol outlines a study aimed at enhancing influenza vaccination uptake among the Chinese older adults through a multifaceted approach combining Information–Motivation–Behavioral Skills (IMB) model-based behavioral interventions and conditional economic incentives. This research addresses the pressing public health issue of low influenza vaccination rates in older adults, who are particularly susceptible to severe complications.

The IMB model, a well-established theoretical framework that has shown promise in altering health behaviors, could provide empirical evidence on the effectiveness of behavioral interventions. The IMB model-based intervention focuses on enhancing older adults’ knowledge of influenza vaccines, motivating them to protect themselves against influenza and improving their behavioral skills to carry out the vaccination act. Interventions guided by this model address various barriers to influenza vaccination, including lack of awareness, insufficient motivation, and obstacles related to accessing vaccination services [33].

This study will further verify whether increasing economic incentives can further improve the effectiveness of the intervention. Economic incentives leverage the power of extrinsic rewards to stimulate behavioral changes, potentially overcoming financial barriers and reinforcing positive vaccination behaviors [41]. The inclusion of economic incentives is an innovative strategy that capitalizes on the potential of monetary rewards to motivate individuals toward health-promoting actions [23]. By analyzing objective vaccination records from community health service center databases and assessing the distinct and combined effects of these interventions, this study will provide valuable insights for shaping policies and practices aimed at promoting influenza vaccination among older adults in China.

Several practical and operational issues should be addressed to ensure the trial’s effectiveness and feasibility. First, to ensure a reasonable response rate and follow-up rate, we will provide participants with clear, easy-to-understand informational materials and ensure that follow-up procedures are streamlined and that participants receive reminders via multiple channels. Second, to ensure the study’s quality, we will provide standard training to all staff involved in participant recruitment and filing work. Third, to establish an objective measure of our primary outcome of vaccination status, we will obtain official vaccination records that will be provided and certified by the local health authorities (instead of self-reported data from our participants). Last, quality control measures will be instituted throughout the whole study process to monitor adherence to protocols and ensure consistency in data collection.

This study has several potential limitations. Firstly, its generalizability may be limited due to its focus on self-paid influenza vaccination within specific communities in China. However, existing research suggests that even in regions with free influenza vaccination, uptake rates remain low [42]. The behavioral interventions and economic incentives explored in this study hold promise for enhancing vaccine acceptance and uptake. Secondly, the effectiveness of economic incentives could be influenced by external factors, such as changes in the financial environment or variations in individual preferences for monetary rewards. To mitigate this bias, we consider affordability as a moderating variable in vaccine uptake. Additionally, given the varying influenza vaccination promotion policies across different regions, the study is subject to overestimating or underestimating the intervention effects. Lastly, it is important to note that the study duration might not fully capture the long-term sustainability of the observed effects. Future research could explore the adaptability of the IMB model and economic incentives in different cultural and economic contexts to provide more generalizable findings.

## 4. Conclusions

This study presents a research design to address the critical issue of low influenza vaccination rates among Chinese older adults. By combining behavioral interventions grounded in the IMB model with conditional economic incentives, we seek to identify effective strategies to significantly improve influenza vaccination uptake in this high-risk population. The findings of this study will have important implications for public health policy and practice, as well as for future research in this field.

## Figures and Tables

**Figure 1 healthcare-12-02361-f001:**
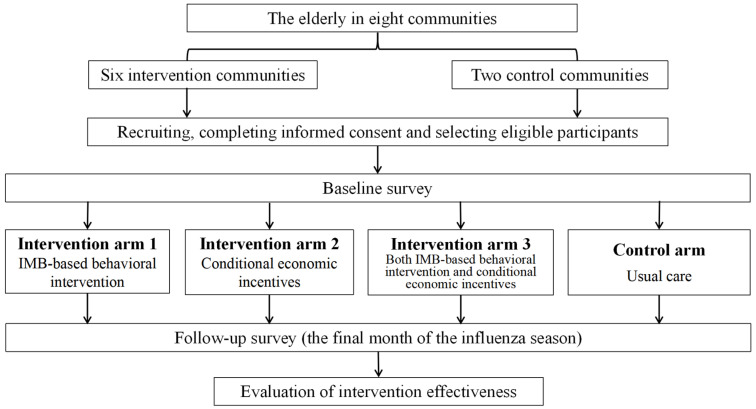
Schematic diagram of study method.

**Figure 2 healthcare-12-02361-f002:**
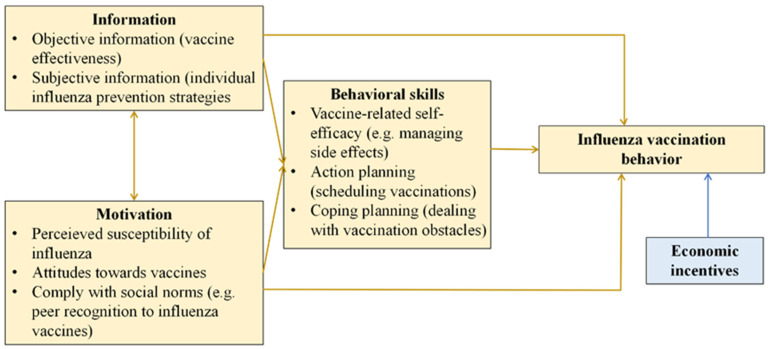
The theoretical framework for influenza vaccination cognition and behavior among older adults in China.

## Data Availability

No new data were created or analyzed in this study.

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
