# Peer review of "Promoting Influenza Vaccination Uptake Among Chinese Older Adults Based on Information–Motivation–Behavioral Skills Model and Conditional Economic Incentive: Protocol for Randomized Controlled Trial"

_healthcare, 2024, doi:10.3390/healthcare12232361_

Round 1
Reviewer 1 Report
Comments and Suggestions for Authors
The rationale for this study protocol stems from existing gaps in knowledge regarding factors that influence elderly individuals' willingness to receive influenza vaccines and related behavioral interventions. The aim of this study protocol is to plan the effectiveness of an IMB (Information-Motivation-Behavioral Skills) model-based behavioral intervention and conditional economic incentives in increasing influenza vaccination rates among the elderly in China through a randomized controlled trial. The literature related to the subject is sufficiently summarized in the protocol.
This study will employ a cluster randomized controlled trial design, with communities as the unit of randomization, to assess the effectiveness of an IMB-based behavioral intervention and conditional economic incentives on influenza vaccination rates among elderly Chinese individuals. Eight communities across urban and rural areas in Hubei and Shandong provinces will be randomly allocated to four groups: IMB intervention, economic incentives, combined intervention and incentives, and control. Influenza vaccine uptake will be measured before and after the interventions across the study groups. The study design is clearly presented in both the figure and the text. Inclusion and exclusion criteria, sample size determination were given appropriately.
Minor comments:
*Line 271: Information should be gathered on the presence of chronic illnesses in the case and pneumococcal vaccination status.
*Line 275-283: It should be specified which reference(s) were used to develop the questions on 'Influenza vaccination intentions and status.
*Line 284-302: ”Influenza vaccine literacy: Participants will answer questions about general vaccine literacy and influenza vaccine literacy.” The development process of these questions and their validation studies should be explained.
*Line 303-312: For 'IMB-based vaccination cognitions,' an explanation should be provided on how the questions were/will be developed, including information on validation, similar studies, and related surveys.
This is a very good protocol, and I am looking forward to its implementation and seeing its effectiveness.
Reviewer 2 Report
Comments and Suggestions for Authors
There are some revisions that I would like the authors to address and/or to consider.
1. Please address and follow the SPIRIT Checklist for developing and presenting a clinical trial study protocol
2. Please don’t use Elders/ elderly / elderly individuals. These words pose ageism. instead of them please use older adults, older people
3. It is essential to justify that remaining low influenza vaccine uptake rates among the Chinese older adults is related to low information, low motivation and low behavioral skills , so that the IMB model is appropriate for this study
4. Discussion section should include and discuss any practical or operational issues involved in performing the study
Round 2
Reviewer 2 Report
Comments and Suggestions for Authors
I would like to thank the authors for considering the comments and changing the manuscript accordingly.